# netDx: interpretable patient classification using integrated patient similarity networks

Shraddha Pai[1,2] iD, Shirley Hui[1], Ruth Isserlin[1], Muhammad A Shah[1], Hussam Kaka[1] &
Gary D Bader[1,3,4,5,*] iD

## Abstract

Patient classification has widespread biomedical and clinical applications, including diagnosis, prognosis, and treatment response prediction. A clinically useful prediction algorithm should be accurate, generalizable, be able to integrate diverse data types, and handle sparse data. A clinical predictor based on genomic data needs to be interpretable to drive hypothesis-driven research into new treatments. We describe netDx, a novel supervised patient classification framework based on patient similarity networks, which meets these criteria. In a cancer survival benchmark dataset integrating up to six data types in four cancer types, netDx significantly outperforms most other machine-learning approaches across most cancer types. Compared to traditional machine-learning-based patient classifiers, netDx results are more interpretable, visualizing the decision boundary in the context of patient similarity space. When patient similarity is defined by pathway-level gene expression, netDx identifies biological pathways important for outcome prediction, as demonstrated in breast cancer and asthma. netDx can serve as a patient classifier and as a tool for discovery of biological features characteristic of disease. We provide a free software implementation of netDx with automation workflows.

**Keywords** multimodal data integration; multi-omics; patient similarity networks; precision medicine; supervised machine learning
**Subject Categories** Computational Biology; Genome-Scale & Integrative Biology; Systems Medicine
**Mol Syst Biol. (2019) 15: e8497**

## Introduction

The goal of precision medicine is to build quantitative models that guide clinical decision-making by predicting disease risk and response to treatment using data measured for an individual. Within the next 5 years, several countries will have general-purpose cohort databases with 10,000 to > 1 million patients, with linked genetics, electronic health records, metabolite status, and detailed clinical phenotyping; examples of projects underway include the UK BioBank (Sudlow *et al*, 2015), the US NIH Precision Medicine Initiative (https://obamawhitehouse.archives.gov/precision-medicine), and the Million Veteran Program (http://www.research.va.gov/MVP/). Additionally, human disease-specific research projects are profiling multiple data types across thousands of individuals, including genetic and genomic assays, brain imaging, behavioral testing, and clinical history from integrated electronic medical records (Hudson *et al*, 2010; Calkins *et al*, 2015; Collins & Varmus, 2015) (e.g., the Cancer Genome Atlas, http://cancergenome.nih.gov/). Computational methods to integrate these diverse patient data for analysis and prediction will aid understanding of disease architecture and promise to provide actionable clinical guidance.

Statistical models that predict disease risk or outcome are in routine clinical use in fields such as cardiology, metabolic disorders, and oncology (Wilson *et al*, 1998; Schmidt *et al*, 2005; Gail *et al*, 2007; Lee *et al*, 2014). Traditional clinical risk prediction models typically use generalized linear regression or survival analysis, in which individual measures are incorporated as terms (or features) of a single equation. Standard methods of this type have limitations analyzing large data from genomic assays (e.g., whole-genome sequencing). Machine-learning methods can handle large data, but are often treated as black boxes that require substantial effort to interpret how specific features contribute to prediction. Black box methods are unlikely to be clinically successful, as physicians frequently must understand the characteristic features of a disease to make a confident diagnosis (Castelvecchi, 2016). Interpretability is particularly required in genomics because of relatively smaller sample sizes and to better understand the molecular causes of disease so that targeted therapies can be designed. Further, many existing methods do not natively handle missing data, requiring data pruning or imputation, and have difficulty integrating multiple different data types.

The patient similarity network framework can overcome these challenges and excels at integrating heterogeneous data and

1 The Donnelly Centre, University of Toronto, Toronto, ON, Canada
2 Affiliate Scientist, The Centre for Addiction and Mental Health, Toronto, ON, Canada
3 Department of Molecular Genetics, University of Toronto, Toronto, ON, Canada
4 Department of Computer Science, University of Toronto, Toronto, ON, Canada
5 The Lunenfeld-Tanenbaum Research Institute, Mount Sinai Hospital, Toronto, ON, Canada
*Corresponding author. E-mail: gary.bader@utoronto.ca

generating interpretable models. In this framework, each feature of patient data (e.g., gene expression profile, age) is represented as a patient similarity network (PSN; Fig 1A). Each PSN node is an individual patient, and an edge between two patients corresponds to pairwise similarity for a given feature. For instance, two patients could be similar in age, mutation status, or transcriptome. PSNs can be constructed based on any available data, using a similarity measure (e.g., Pearson correlation and Jaccard index). Because all data are converted to a single type of input (similarity networks), integration across diverse data types is straightforward. Patient similarity networks (PSN) are a recently introduced concept and have been used successfully for unsupervised class discovery in cancer and type 2 diabetes (Wang *et al*, 2014; Li *et al*, 2015), but have never been developed for supervised patient classification.

We describe netDx, the first PSN-based approach for supervised patient classification. In this system, patients of unknown status can be classified based on their similarity to patients with known status. It applies the idea of recommender systems, similar to those used in Amazon or Netflix ("find movies like this one"), to precision medicine ("find patients who don't respond to therapy"). This process is clinically intuitive because it is analogous to clinical diagnosis, which often involves a physician relating a patient to a mental database of similar patients they have seen. As demonstrated below, netDx has strengths in classification performance, heterogeneous data integration, and interpretability.

# Results

## Algorithm description

The overall netDx workflow is shown in Fig 1, with detailed views of particular steps in Figs EV1 and EV2. The example in Fig 1 conceptually shows how patient similarity networks (PSNs) can be used to predict whether a patient is at high or low risk of developing a disease based on one or more patient-level data types. Similarity networks are computed for each patient pair and for each data type. In this example, high-risk patients are more strongly connected based on their clinical profile, which may capture age and smoking status, and metabolomic profile. Low-risk patients are more similar in their clinical and genomic profiles. The goal of netDx is to identify the input features predictive of high and low risk, and to accurately assign new patients to the correct class.

### Input data design

Each patient similarity network (PSN) is a feature, similar to a variable in a regression model. A PSN can be generated from any kind of patient data, using a pairwise patient similarity measure (Fig 1A and B). For example, gene expression profile similarity can be measured using Pearson correlation, while patient age similarity can be measured as normalized age similarity. A reasonable design is to define one similarity network per data type, such as a single network based on correlating the expression of all genes in the transcriptome, or a single network based on similarity of responses to a clinical questionnaire. If a data type is multivariate, defining a network for each individual variable will result in more interpretable output. However, this approach may lead to too many features generated (e.g., one network each for millions of SNPs), which increases

computational resource requirements and risk of overfitting. Thus, as with any machine-learning task, there is a trade-off between interpretability and overfitting/scalability. To help address this problem for gene-oriented data (e.g., transcriptomics), we group gene-based measurements into biological pathways, which we assume capture relevant aspects of cellular and physiological processes underlying disease and normal phenotypes. This biological process-based design generates ~2,000 features from gene expression profiles containing over 20,000 genes, with one feature per pathway, but has the limitation that not all genes can be mapped to pathways.

### Selecting features informative for class prediction

In machine learning, the feature selection step identifies the features (here, networks) with the highest generalizable predictive power. In the case of netDx, predictive features are identified for each class using feature selection. netDx is trained on samples from the class of interest, using cross-validation (Figs EV1 and EV2) and an established network integration algorithm (GeneMANIA; Mostafavi *et al*, 2008). When provided with a collection of PSNs, GeneMANIA takes a query of patients as input ("+" nodes) and solves a constrained regression problem on the network to maximize edges that connect query patients, that is, enriched for (+,+) interactions, relative to other edges in the network. The ideal network is one connecting all patients of the same class without any connections to other classes; an example would be a network connecting all treatment responders, and all non-responders, without edges between the two. The least useful network is one that connects patients from one class to patients from other classes, without connecting any patients within the same class. The query-driven regression weights each network, with higher weights reflecting a greater enrichment of (+,+) edges. netDx assesses the robustness of this feature selection by performing repeated GeneMANIA queries with different subsamples of the training samples. The netDx score for a given feature is the number of times that feature was assigned a positive score in a query across resampling rounds. This scoring process is repeated for each class. The classifier's sensitivity and specificity can be tuned by thresholding this score (for instance, features passing selection could be defined as those scoring $\geq 80\%$); a feature with a higher score achieves greater specificity and lower sensitivity. Similar to feature selection in other machine-learning algorithms, the output of this step is a set of features that can be integrated to produce a predictor for the patient class of interest.

### Class prediction using selected features

Selected features are combined by averaging their similarity scores to produce a single, integrated network for each class. GeneMANIA runs label propagation on each integrated network outward from the query nodes ("+" nodes) to rank all other nodes in the network by similarity to the query (Fig EV2). For each class, netDx generates a GeneMANIA database consisting of networks passing feature selection; for example, one database for high-risk of lung cancer and another for low-risk of lung cancer. It then runs a single Gene-MANIA query on this database for each class, using the training samples as the query. This is equivalent to the query, "given predictive networks for patient class L, rank all patients by similarity to known examples of L". netDx then assigns test patients to the class for which it has the highest similarity (Mostafavi *et al*, 2008; Figs EV1 and EV2).

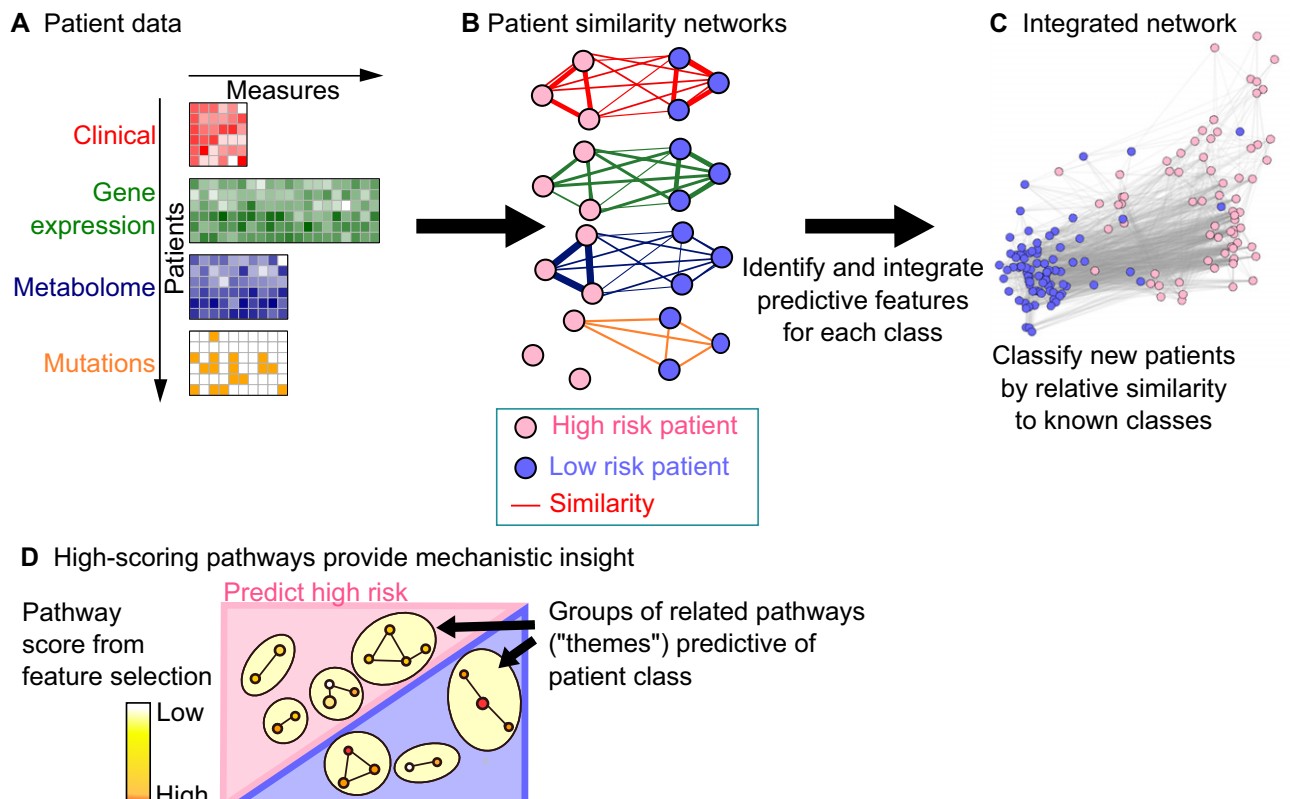

**A** Patient data

**B** Patient similarity networks

**C** Integrated network

Identify and integrate predictive features for each class

Classify new patients by relative similarity to known classes

○ High risk patient
● Low risk patient
— Similarity

**D**  High-scoring pathways provide mechanistic insight

Predict high risk

Pathway score from feature selection

Low

High

Groups of related pathways ("themes") predictive of patient class

Predict low risk

**Figure 1.  The netDx method.**

A   Patient data, provided as input to netDx in the form of tables. The simple example for predicting low/high risk of disease uses clinical, genomic, metabolomic, and genetic data as input.

B   Patient similarity networks (or PSN) are networks with patients as nodes and weighted edges representing patient similarity by some measure. netDx converts patient data into a set of patient similarity networks. netDx identifies which networks strongly relate high-risk patients (here, clinical and metabolomic data) and which relate low-risk patients (clinical and gene expression data). Feature selection is used to score networks by their ability to predict patient class (details in Fig EV1).

C   netDx output. netDx returns several types of output. Top-scoring features are combined into a single view, or network of overall patient similarity, which can be used to classify new patients based on relative similarity to known patient classes. netDx also provides standard classifier performance metrics and scores for the predictive value of individual features.

D   Network-based visualization of top predictive pathways. If pathway features are used, netDx identifies and visualizes the pathways most useful for classification.

*netDx output*

netDx returns the set of selected features for each class, predicted labels for all test patients, and standard performance measures including the area under the receiver operating characteristic curve (AUROC), area under the precision-recall curve (AUPR), and accuracy (Fig 1C). Class-specific scores for each feature are returned; if pathway features are used, they are visualized using an enrichment map (Fig 1D; Merico *et al*, 2011a). A final integrated patient similarity network is generated by integrating all selected features. Inter- and intra-class separation is measured using average shortest path methods over the integrated PSN (Fig 1C), and the network is visualized to aid interpretation of the strength of class separation.

**Benchmarking performance by predicting binarized survival in cancer**

To assess the classification performance of netDx, we use an established cancer survival prediction benchmark available for four tumor types, using data from The Cancer Genome Atlas (TCGA; http://

cancergenome.nih.gov/) via the TCGA PanCancer Survival Prediction project website of Yuan *et al* (2014a), https://www.synapse.org/#! Synapse:syn1710282). These tumor types have been thoroughly analyzed using eight machine-learning methods, which provide extensive performance results that we can compare to (Yuan *et al*, 2014a). Data are for renal clear cell carcinoma (The Cancer Genome Atlas, 2013; KIRC, $N = 150$ patients; Data ref: Yuan *et al*, 2014b), ovarian serous cystadenocarcinoma (The Cancer Genome Atlas, 2011; OV, $N = 252$, Data ref: Yuan *et al*, 2014c), glioblastoma multiforme (The Cancer Genome Atlas, 2008; GBM, $N = 155$, Data ref: Yuan *et al*, 2014d), and lung squamous cell carcinoma (The Cancer Genome Atlas, 2012a; LUSC, $N = 77$, Data ref: Yuan *et al*, 2014e). Data for a given tumor type include clinical variables (e.g., age and tumor grade); mRNA, miRNA, and protein expression; DNA methylation; and somatic copy number aberrations. Binarization of survival and format of clinical variables followed previous work (Yuan *et al*, 2014a; Data ref: Yuan *et al*, 2014f); comparison of data processing and predictor methods to that of the PanCancer Survival project is shown in Fig EV3 and Appendix Table S1.

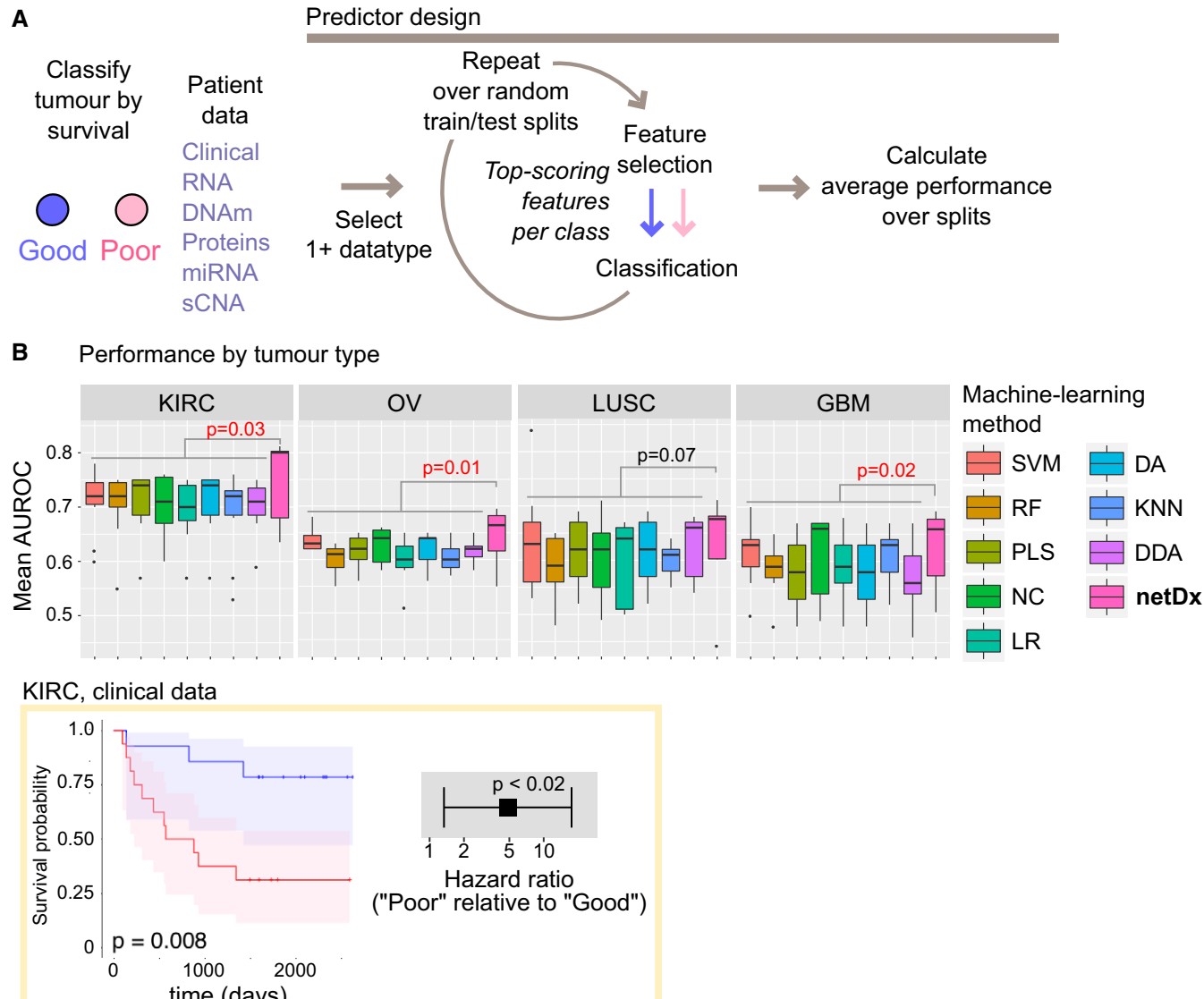

**Figure 2. Performance benchmarking with PanCancer Survival data.**

A  Method. Various combinations of patient data types were provided as input to netDx, to predict binary survival (Good/Poor). Performance was compared for renal clear cell carcinoma (KIRC, N = 150 patients), ovarian cystadenocarcinoma (OV, N = 252), lung squamous carcinoma (LUSC, N = 77), and glioblastoma multiforme (GBM, N = 155).

B  Average performance of netDx compared to other machine-learning methods. Each panel shows data for one tumor type, and each boxplot shows mean AUROC (20 train/test splits) for a given machine-learning method across different tested combinations of input data (Appendix Table S1–S3). Boxplot center indicates median; box bounds indicate 25th and 75th percentile, and whiskers mark 1.5 times the interquartile range. Dots indicate points that fall outside this range. netDx is shown in pink. Statistical significance is computed using a one-sided Wilcoxon–Mann–Whitney comparing netDx to all other methods combined; this was done to establish strictly higher performance of netDx relative to other methods. Bottom: Yellow box: As a reference point, Kaplan–Meier curves and hazard ratios are shown for predicted samples from a representative KIRC split (split with AUROC closest to the mean AUROC across 20 splits). The light shaded pink and blue areas in the Kaplan–Meier curve indicate 95% confidence intervals based on log-hazard; P-value from log-rank test. Error bars for the hazard ratio indicate 95% confidence intervals; P-value from Cox proportional hazards model.

For each tumor type, we classified samples into high and low survival categories using a range of models, each with different combinations of input data, following (Yuan *et al*, 2014a) for comparability (Fig 2A). Each model was independently trained (80:20 train:test) and optimized. For a given train/test split, feature selection was performed over 10 resamplings of the training samples for each class, and features robustly identified in at least 9 out of 10 resamplings were used to classify test samples. This process was repeated for multiple random splits of train and test until performance measures stabilized (20 splits; Fig 2A). Performance for each model was measured as the average of test classification (mean AUROC) across the 20 splits. Optimization of each model included varying choice of similarity metric, whether features were defined at the level of entire data types or of individual variables (genes), and by whether or not imputation was used (Appendix Table S1–S3). Where features were defined at the level of individual genes,

variables were prefiltered using lasso regression to reduce the number of features considered. Prefiltering supplements the robust feature selection step described above and is performed within the resampling loop, by creating features only using selected variables. Appendix Fig S1 shows an example of variability produced in prefiltered features. For similarity metrics, we tested normalized similarity, Pearson correlation with and without exponential scaling, radial basis function, and Euclidean-distance-based similarity with exponential scaling. In total, 40 different models were trained, 9–11 per cancer type (Dataset EV1).

We use the AUROC to measure performance as this was the metric reported by the PanCancer Survival project (Yuan *et al*, 2014a; Appendix Table S3). Information on the exact samples used for the 10 train/test splits used by Yuan *et al* is not available, thus we used new random splits, but used more splits ($n = 20$) until we reached a stable AUROC estimate. For all tumor types, netDx demonstrates performance consistently better than, or at par with, other machine-learning methods (Fig 2B; Dataset EV1). Average netDx performance is significantly higher than that for all other methods for three of the tumors (one-tailed WMW to test strictly higher netDx performance; KIRC: $P = 0.03$; OV: $P = 0.02$; GBM: $P = 0.02$) and is not significant for the fourth (LUSC: $P = 0.07$). Furthermore, the top-performing netDx model outperforms all eight tested machine-learning algorithms for kidney, ovarian, and brain cancer (Fig 2B, Appendix Table S1), and outperforms all but one outlier model for lung cancer (netDx best = 0.72, Yuan *et al* second best = 0.71). Pairwise comparison of netDx broken down by machine-learning method shows that netDx has a significantly better median AUROC score in most (75%) cases (Appendix Table S2). Performance statistics reported by Yuan *et al* were the best performing models out of 320 tested for different data combinations with eight different machine-learning methods: diagonal discriminant analysis; K-nearest neighbor; discriminant analysis; logistic regression; nearest centroid; partial least squares; random forest; and support vector machine. Thus, netDx can perform as well as, or better than, a diverse panel of machine-learning methods.

## Pathway-level feature selection identifies cellular processes predictive of clinical condition

Creating a single feature per data type identifies the general predictive value of that data layer but, without further work, does not provide insight into which genes or cellular processes are useful for classification. To better understand disease mechanisms, netDx supports the ability to group gene-level measures into pathways (gene sets) so that the predictive value of pathway features can be measured. To illustrate this ability, we classified breast tumors as being of the Luminal A subtype or not, using tumor-derived gene expression ($N = 348$ patients; 154 Luminal A, 194 non-Luminal A; The Cancer Genome Atlas, 2012b; Data ref: The Cancer Genome Atlas, 2012c,d). Gene expression was used as input, and features were defined at the level of curated pathways of cellular processes (i.e., 2,119 patient similarity networks, one per pathway; Data ref: Merico *et al*, 2011b). We split the data into train and test groups in an 80:20 ratio and ran netDx as described in the previous section to classify Luminal A vs. "other". This process was repeated with 100 train/test splits to achieve stable pathway-level scores (Appendix Fig S2). Consistently predictive pathways, representing a high-confidence set of features, are visualized as an EnrichmentMap (Fig 3),

with each pathway associated with a consistency score (highest netDx feature selection score in $\geq 70\%$ of the 100 splits).

Classification performance was high, with an average (SD) AUROC of $0.97 \pm 0.01$, average AUPR of $0.92 \pm 0.02$, and average accuracy of $89 \pm 3\%$ (100 train/test splits; Fig 3A). Performance for pathway-level features is slightly, but significantly, better than when gene expression is provided as a single feature (mean AUROC for single network = $0.96 \pm 0.02$; one-sided Wilcoxon–Mann–Whitney test $P = 0.013$). Top-scoring pathways included themes of cell cycle progression and checkpoint regulation, DNA synthesis, DNA mismatch repair, and DNA double-strand break repair (Fig 3B, Dataset EV2). These processes are consistent with the pathways known to be dysregulated in luminal breast tumors and cancer progression in general. netDx also identified pathways related to solute carrier family membrane transport proteins and vesicle release, which are not traditionally linked to breast cancer, but which may support new insights (see Discussion). We integrated the features from top-scoring pathways (those scoring 10 out of 10 in all 100 train/test splits) into a single network (Fig 3C). In this network, LumA patients are significantly closer to other LumA patients (average shortest distance = 0.52), compared to patients of other breast cancer subtypes (average shortest distance = 0.59; one-tailed Wilcoxon–Mann–Whitney test $P < 2e-16$). Therefore, top-scoring features succeed in separating same-class patients, relative to patients of other classes.

A common problem with genomic data is relating cohort-level analysis results, such as the set of affected pathways in Fig 3B, to changes in individual patients. To address this, we performed principal component analysis on gene expression values of individual selected pathways and correlated the projections of the first three principal components with clinical outcome (Fig 3). Most features individually showed significant correlation with tumor subtype (e.g., correlation for "Amplification of Signal from the Kinetochores" = $-0.80$, *t*-test, $P = 3.3e-72$), and the patient class boundary is visually evident in these features (Fig 3D). However, not all features had this property (e.g., correlation for "Glucuronidation" = 0.1, *t*-test, $P = 0.038$). Pathways that score highly in feature selection and correlate with outcome are good candidates for follow-up biomarker or mechanistic studies.

As a second case study to demonstrate that netDx identifies pathways consistent with the biology of the condition, we predicted case/control status in asthma using gene expression from sorted peripheral blood mononuclear cells (Yang *et al*, 2015; 97 cases, 97 controls; Data ref: Yang *et al*, 2015) using an identical predictor design as used for breast cancer above (2,119 pathway-level features). The netDx predictor achieved an AUROC of $0.70 \pm 0.07$ (SD) (Fig 3E; mean AUPR = 0.65; mean accuracy = 66%). Feature-selected pathways included cytotoxic T-lymphocyte-related processes and Notch2 signaling (Fig 3D; Appendix Table S4). These themes are consistent with prior knowledge of cellular changes in asthma (see Discussion). Similar to the breast cancer example, pathway-level features result in significantly improved classification performance, relative to a predictor design where gene expression is provided as a single feature (100 train/test splits; mean AUROC = 0.56, AUPR = 0.54; one-sided WMW for greater performance of pathway-based design, $P < 2e-16$). These examples demonstrate that when used with pathway-level features, netDx can provide insight into molecular mechanisms and disease-related processes that discriminate patient groups. Altogether, our results

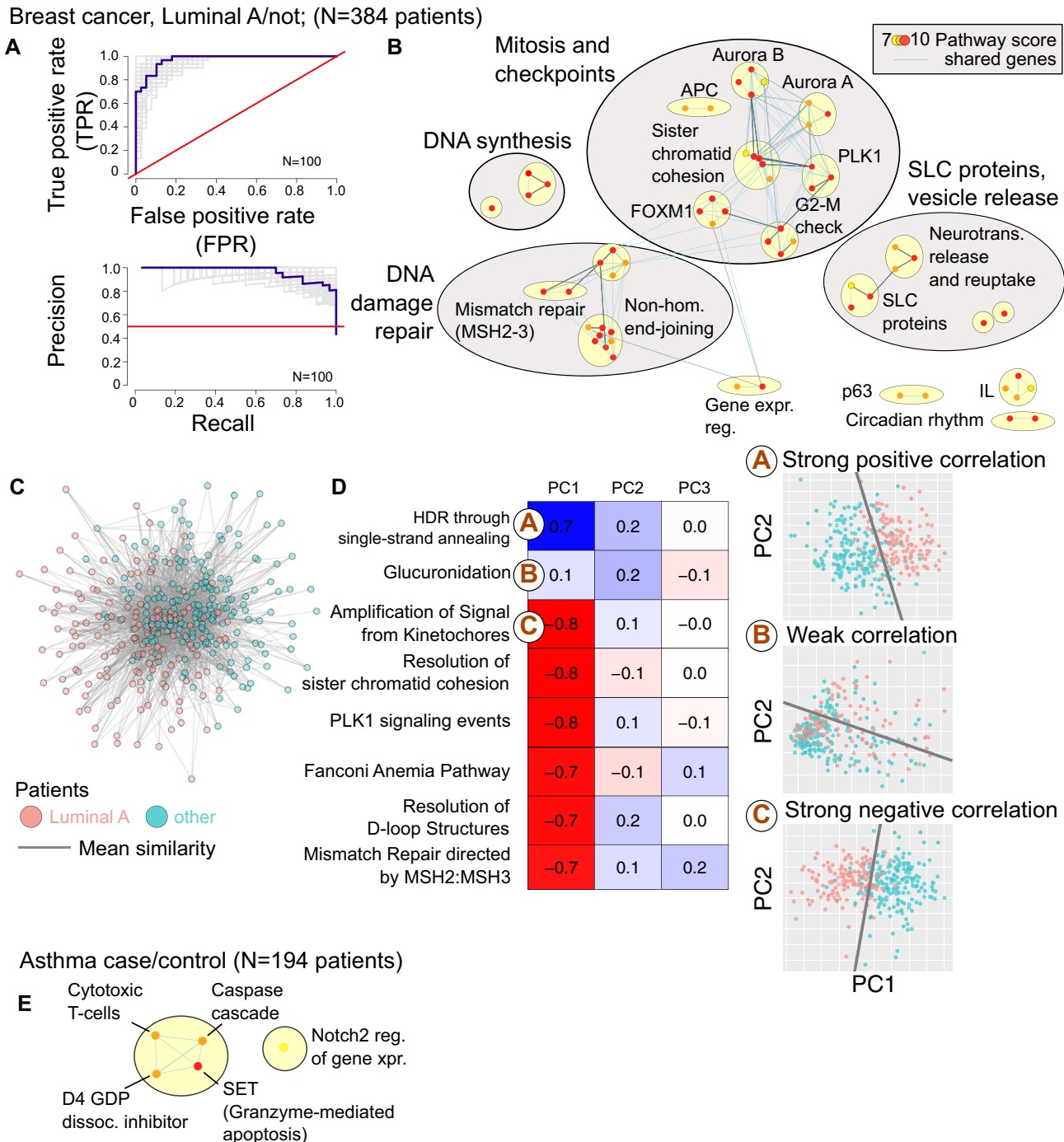

Figure 3. Pathway-level feature selection in breast cancer and asthma.

A    netDx performance for binary classification of breast tumor as Luminal A subtype from tumor-derived gene expression (N = 384 patients).

B    Pathways feature-selected by netDx in predicting Luminal A status. Nodes are pathways, and edges indicate shared genes. Nodes are colored by highest netDx score consistently achieved out of a maximum possible of 10, in ≥ 70% of 100 train/test splits. Themes identified by AutoAnnotate (Merico et al, 2011a; Kucera et al, 2016).

C    Integrated patient similarity network. Nodes represent patients, and edges represent average similarity computed from pathways that scored 10 out of 10 in all splits. Nodes are colored by tumor type. Edges with weight < 0.7 were excluded, and the top 20% of edges per node were retained. The resulting network was visualized in Cytoscape (spring-embedded layout, spring strength = 5).

D    Correlation of top-scoring pathway features (represented as the first three principal components of pathway-specific gene expression) with tumor type (Spearman's correlation). Table cells are colored by sign and magnitude of correlation (blue: Spearman corr. > 0; red, corr. < 0). Circled letters correspond to detailed panels on the right. Right: Projections of patient-level gene expression in feature-selected pathways onto the first two principal components (individual dots indicate patients). Points are colored by survival class. Decision boundaries were calculated using logistic regression on scatterplot data.

E    Selected features for asthma case status in the case of asthma case/control prediction (N = 97 cases; N = 97 controls). Legend as in (B).

show that using pathway-level features can improve classification performance and provide insight into disease mechanisms.

## Discussion

We describe netDx, the first supervised patient classification system based on patient similarity networks. We demonstrate that netDx does as well as or better than a diverse panel of machine-learning approaches in predicting survival across four different tumor types. Further, feature selection, especially when biological pathways are used, aids interpretability and provides insight into disease mechanisms important for classification. This framework can be used to create accurate, generalizable predictors and has particular strengths in data integration and interpretation compared to standard machine-learning approaches. netDx is targeted at researchers who are interested to see whether their sample-level data can answer a specific sample classification question. netDx provides a standard workflow that can determine whether the given classification question can be answered based on a training set and if so, provides a set of relevant features and a software tool to classify new samples.

With the PanCancer benchmark set, netDx performs as well as or better than all machine-learning methods tested. In the case of KIRC, the basic netDx model outperformed other methods without any additional tuning. Some of this performance gain may be due to the use of regularized regression in GeneMANIA. In the context of predicting gene function, the original GeneMANIA benchmarking tests determined that, relative to unregularized linear regression, regularization improved performance by reducing overfitting (Mostafavi *et al*, 2008). In other instances (LUSC, OV, and some GBM models), tuning the sparsification level of the input networks changed netDx's performance sufficiently to outperform other methods. Parameters to alter sparsification included how many of each patient's strongest edges were retained (i.e., top 30–50 edges per patient), and whether or not an exponential scaling filter was applied to edge weights. We found that the optimal parameter choice differed depending on the dataset (Dataset EV1). Similar to regularization, the sparsification of input networks was demonstrated to improve GeneMANIA performance in gene function prediction, presumably by accentuating signal and reducing noise (Mostafavi *et al*, 2008). We therefore attribute the improved performance of netDx to the fine tuning of the input networks through sparsification and to the use of regularized regression in the GeneMANIA algorithm used by netDx. A single support vector machine model from Yuan *et al* for lung cancer survival prediction vastly outperformed any other model, including netDx, perhaps because it identified a useful non-linear decision boundary, or was overfit. Future work will explore whether considering non-linear effects (e.g., via non-linear similarity measure and network combinations) can improve performance.

netDx includes support for grouping genes into pathways to improve interpretability. Grouping genes into a smaller number of features can mitigate overfitting risk, improve signal detection with sparse data, be easier to analyze, and improve prediction performance. The themes identified for Luminal A classification of breast tumors are consistent with processes known to be dysregulated in this type of cancer. For instance, themes of DNA repair and G2-M checkpoint regulation are consistent with the known roles of BRCA1/BRCA2 and ATM proteins, which are established risk factors for breast cancer

(Roy *et al*, 2011). Cell cycle dysregulation accompanies genomic instability as a feature of several cancers (Negrini *et al*, 2010). netDx also identified a theme of solute carrier family proteins, many of which are overexpressed in tumors, are thought to support the metabolic needs of growing tumors (McCracken & Edinger, 2013; Uhlen *et al*, 2015) and are associated with genetic risk of breast cancer (Fletcher *et al*, 2011). These suggest novel directions for biomarker identification and therapeutic targeting. Similarly, the themes identified for asthma case prediction, including cytotoxic T lymphocytes and associated apoptosis, are consistent with known asthma genetics and genomic results. netDx also identified Notch signaling as predictive of asthma. Notch signaling regulates the differentiation of T-helper cells, and inhibitors of this pathway are being tested in clinical trials to suppress symptoms of asthma (Okamoto *et al*, 2008, 2009; Huang, *et al*, 2017). In summary, when provided with pathway-level features, netDx can be a useful tool for discovery research.

We compared the output of netDx's pathway-based patient classifier to traditional pathway enrichment analysis (gene set enrichment analysis or GSEA; Subramanian *et al*, 2005), by applying GSEA to the Luminal A vs. "other" gene expression comparison data mentioned previously ($N = 348$ patients, expression for 17,814 genes). GSEA identified roughly ~1.5 times more pathways (126 pathways, $Q < 0.05$; Dataset EV3) than netDx did (80 pathways scoring $\geq 7$ out of 10 in $\geq 70$ out of 100 train/test splits). Roughly half of netDx identified pathways (48%) were also found by GSEA (Appendix Fig S3, Dataset EV4), and these overlapping pathways are all known to be affected in breast cancer (e.g., cell division, cell cycle checkpoints, DNA replication, and DNA damage repair). Overall, netDx is more conservative than GSEA in terms of pathways reported, likely because netDx uses regularized regression to reduce redundancy in the feature selection step.

We also compared netDx to the DIABLO multi-omic patient classifier (Rohart *et al*, 2017) which uses a partial latent structure-based method to identify correlation between individual features, such as gene expression levels, and with outcome. For this, we integrated mRNA and miRNA expression from 346 primary breast tumors (The Cancer Genome Atlas, 2012b), using pathway-level features in netDx and, for comparison, the set of genes covered by these pathways in DIABLO. Both methods used identical train/test splits of 80:20. Both methods performed well (AUROC: DIABLO: average AUROC = 0.86; netDx: 0.96; accuracy: DIABLO: 90%, netDx 91.4%; Dataset EV5 and EV6). netDx selected pathways with themes similar to the Luminal A gene expression-based predictor previously mentioned (Fig 3B), including cell cycle regulation and DNA damage repair (Appendix Fig S4A). DIABLO identified a range of features correlated with outcome, such as the known link of *mir-30a* overexpression with improved survival in both luminal and basal tumors (Kawaguchi *et al*, 2017). It also highlighted individual genes present in the pathways found by netDx (Appendix Fig S4B), for example, *CENPA* and *AURKB* in Aurora signaling pathways. DIABLO focuses on gene-level features, and netDx natively supports pathway-level features; thus, both tools provide complementary views of predictive multi-omic features that could be useful when applied in tandem.

In conclusion, netDx is a useful tool for precision medicine because it combines a conceptually intuitive paradigm of patient similarity networks with classification performance better than or equal to traditional machine-learning methods, as well as biological interpretability when using pathway features. Like other machine-learning methods,

model performance and generalizability are limited by feature design choices and sample size. Our ultimate vision is to enable clinical researchers to assess classification performance for questions of interest, such as "will a patient respond to one therapy or another?" based on patient measurements and outcomes present in large electronic medical record databases. Output would include a report card on model performance and generalizability estimates on independent cohorts, feature interpretation, an interactive integrated patient similarity network visualization enabling the exploration of individual patients, and a ready-to-run classifier for new patients (Pai & Bader, 2018).

One limitation of netDx, as for any supervised machine-learning method, is the need for a large bank of patient samples to learn from. Fortunately, clinical data are growing rapidly, though specific queries involving smaller patient cohorts will need alternative methods to address. Additionally, netDx needs to compute similarities of training as well as test patients. Private data will therefore potentially need to be shared with users of any netDx classifier. Secure sharing, differential privacy, or privacy-preserving data mining techniques could be developed to enable patient similarity computations without the need to share complete private data. We therefore expect that netDx will initially be useful as a multi-omic data analysis tool for research groups rather than in a production clinical environment.

netDx is implemented as an open-source R software package available at http://netdx.org, with worked examples and a Docker container enabling reproduction of the results and figures in this paper (https://doi.org/10.5281/zenodo.2558452). We also propose that users store and publicly share patient similarity networks, useful as features for netDx and other PSN methods, in the NDEx network exchange system (Pratt *et al*, 2015).

# Materials and Methods

## PanCancer Survival benchmark models

We tested various models for the PanCancer survival benchmarking. This section describes the model details; models are named as per Dataset EV1. The models varied based on whether or not they included a data imputation step, whether or not variables were prefiltered using lasso regression, and choice of similarity metric (Pearson correlation, normalized similarity, scaled Euclidean/Pearson). Where used, imputation or prefiltering was performed only on training samples inside the cross-validation (CV) loop to avoid leaking information from train to test.

### Base (no lasso prefiltering)
In this model, each data type was treated as a single feature (e.g., one patient similarity network was generated for gene expression, one for clinical data). Similarity was defined by Pearson correlation where a data type had more than six measures (Abdel-Megeed, 1984) or by average normalized similarity if the data type had five or fewer variables. For the case of a single continuous variable, we use the normalized similarity, defined as follows:

$$S(a, b, G) = 1 - \frac{abs(a - b)}{max(G) - min(G)}$$

Where $a$ and $b$ are the values of the variable for individual patients ($a$ and $b$) and $G$ is the set of all values for the variable (e.g., age). For a set of $k$ variables G={$g_1$,$g_2$,..$g_k$}, where $1 \le k \le 5$, the similarity $S'$ between two patients $a$ and $b$ is defined as the average of normalized similarity for each of the variables:

$$S'(a, b, G) = \frac{\sum_{i=1}^{k} S(a, b, g_i)}{k}$$

### Variable prefiltering and scaled Euclidean/scaled Pearson similarity metrics
This model design defines features at the level of individual variables (e.g., genes and clinical variables) and performs within-CV prefiltering using lasso regression to rank features by their ability to predict outcome (Tibshirani, 1996). This method is likely a better choice than combining all variables of a data type into a single network when only a handful of variables carry predictive information for that data type. Only variables with a non-zero weight are included in the analysis. Regression uses only training samples within a given fold to avoid leaking information from train to test. The similarity metric used is either Euclidean distance (model code = *eucscale*) or Pearson correlation, followed by local exponential scaling (Wang *et al*, 2014). Imputation (*pearimpute*, *eucimpute*) is performed within the cross-validation loop separately for training and test samples to avoid leaking information from train to test. The lung cancer dataset demonstrated the best performance if the model was also limited to the top clinical variable from lasso (*plassoc1*).

### Integrated patient network

The integrated patient network is an average combination of all selected networks (features) to create a single network (i.e., average of all edge weights between patients from all selected networks). Visually, the goal is to view similar patients as being more tightly grouped and dissimilar patients as being farther apart. Similarity (normalized from 0 to 1) is therefore converted to dissimilarity, defined as 1-similarity. Weighted shortest path distances are computed on this resulting dissimilarity network. The one-tailed Wilcoxon–Mann–Whitney test is used to ascertain whether within-class distances are collectively shorter than across-class distances. To aid visualization, only edges representing the top 40% of distances in the network are included. The weighted shortest path between patient classes (a node set) in the integrated network was computed using Dijkstra's method (*igraph* v1.01; Csardi & Nepusz, 2006); distance was defined as 1-similarity (or edge weight from a patient similarity network). The overall shortest path was defined as the mean pairwise shortest path for a node set.

### Survival curve and hazard ratios

Survival curves were constructed based on netDx-predicted classes of test samples. The R packages *survival* and *survminer* were used to compute Kaplan–Meier curves, and *rms* was used to calculate the log-rank test for separation of survival curves. The package *survival* was also used to compute the Cox proportional hazards model of predicted poor survivors, using predicted good survivors as a reference, and to calculate the hazard ratio and associated *P*-value.

## Pathway networks

Pathway definitions were aggregated from HumanCyc (Romero *et al*, 2005; http://humancyc.org), NetPath (Kandasamy *et al*, 2010; http://www.netpath.org), Reactome (Croft *et al*, 2014; Fabregat *et al*, 2016; http://www.reactome.org), NCI Curated Pathways (Schaefer *et al*, 2009), mSigDB (Subramanian *et al*, 2005; http://software.broadinstitute.org/gsea/msigdb/), and Panther (Mi *et al*, 2005; http://pantherdb.org/). The compiled set of pathways was downloaded from http://download.baderlab.org/EM_Genesets/February_01_2018/Human/symbol/Human_AllPathways_February_01_2018_symbol.gmt (Merico *et al*, 2011a). Only pathways with 10–200 genes were included (2,119 pathways). Pathway-level patient similarity was defined as the Pearson correlation of the expression vectors corresponding to member genes, and the network was sparsified (see next section).

For the asthma case study, Ensembl IDs were mapped to HGNC symbols using the Bioconductor package org.Hs.eg.db.

## Sparsification of input networks

Sparsification is useful for reducing the number of edges to be considered in input networks, speeding computation, and for reducing noise in the data, as weak correlations are removed. Sparsification uses three parameters: the minimum edge weight to include (*cutoff*; set at smallest possible non-zero weight for models using exponential scaling, and at 0.3 for others), how many top interactions to include per node (*topX*), and the upper bound on the number of edges in a network (*maxEdges*). By default, GeneMANIA sparsifies networks by first excluding edge weights below a cutoff and retaining the top 50 edges per node (Mostafavi *et al*, 2008). We used this sparsification strategy in all instances where Pearson correlation was used to compute patient similarity. For other similarity measures (e.g., normalized similarity) or where exponential scaling was applied to networks, we found that additionally limiting the total number of edges in the network (i.e., adding a *maxEdges* parameter) improved performance in some models. In the latter instance, we ensured that a network included all input patients (so test patients could be classified) by adding the strongest edge for any patients excluded by the *maxEdge* criterion. If this edge weight was lower than the network cutoff, it was set to be at the cutoff value. Different tumor datasets performed optimally with different levels of sparsification: KIRC: *topX* = 30, *maxEdges* = 6,000; LUSC: *topX* = 40, *maxEdges* = 6,000; GBM: *topX* = 50, *maxEdges* = 3,000.

## Map of selected networks

The Enrichment Map app (3.1.0RC4) in Cytoscape 3.6.1 (Shannon *et al*, 2003) was used to generate enrichment maps (Merico *et al*, 2011a). A Jaccard overlap threshold of 0.05 was used to prune identical gene sets. AutoAnnotate v1.1.0 was used to cluster similar pathways using MCL clustering with default parameters.

## Gene set enrichment analysis for LumA subtype in breast cancer

*Limma* (Linear Models for Microarray and RNA-Seq Data; Ritchie *et al*, 2015) was used to fit a linear model to compare mRNA levels of LumA breast cancer vs. other breast cancer subtypes (i.e., Her2,

Basal, LumB, and Normal) in the TCGA BRCA mRNA microarray dataset. This analysis produced a list of genes ranked by a moderated *t*-statistic. As we were interested in both up- and down-regulated genes (i.e., overall change in LumA and other subtypes taken together), we ranked the limma output based on the absolute value of the moderated *t*-statistic. A weighted pre-ranked GSEA (gene set enrichment analysis; Subramanian *et al*, 2005) analysis was performed using the ranked list of genes obtained from the above limma analysis as input. Gene sets were filtered out if they had < 10 or > 500 genes and 1,000 permutations were run. The set of pathway definitions was identical to that used in the netDx breast cancer predictor.

An Enrichment Map was created using the union of GSEA pathways enriched in LumA and other subtypes (FDR < 0.05) and netDx pathways that consistently had high scores across multiple train/test splits (maxScore ≥ 7). This enrichment map was created to assess overlaps in pathways and themes (where a theme is a group of related pathways). We used the Enrichment Map 3.1.0 app (Merico *et al*, 2011a) in Cytoscape 3.6.1 (Shannon *et al*, 2003) and the GMT file Human_AllPathways_February_01_2018_symbol.gmt (same as that used for the pathway-based netDx breast cancer tumor subtype predictor).

## Comparison of netDx and DIABLO for multi-omic classification

Previously published gene expression and miRNA data for 346 tumors were downloaded from the Cancer Genome Atlas data portal (https://tcga-data.nci.nih.gov/docs/publications/brca_2012/; BRCA.348.precursor.txt; BRCA.exp.348.med.txt; The Cancer Genome Atlas, 2012b).

miRNA were mapped to pathways. The MSigDB miRNA gene set database (c3.mir.v6.2.symbols.gmt) (Subramanian *et al*, 2005) was used to map individual miRNA to genes, and the pathway gene set database (Human_AllPathways_February_01_2018_symbol.gmt) was used to assign each miRNA to pathways in which it had a target gene. For comparison to netDx, we limited RNA and miRNA variables provided as input to DIABLO to those present in pathways. Both methods also used identical train/test splits.

For netDx, a predictor was run with pathway-level features for a single train/test split of 80:20. Only pathways with 10–200 genes (2,119 pathways) or those with more than 5 miRNA (for stable computation of Pearson correlation, after (Abdel-Megeed, 1984)) (1,935 pathways) were included. Pathway-level patient similarity was defined by Pearson correlation (network sparsification parameters: edges with correlation below 0.3 were ignored, top 50 strongest edges per node were retained, to a maximum of 3,000 edges per network). RNA and miRNA patient similarity networks were constructed using the same parameters. Training samples were used to construct pathway-level features (4,054 features), and these were scored between 0 and 10 for predictive value, as with previous netDx predictors; features with score ≥ 9 were used to classify the blind test samples.

We ran DIABLO with a gene-centered model, following operating procedures specified in the online tutorial (http://mixomics.org/mixdiablo/case-study-tcga/). The design matrix was a 2 × 2 matrix with 0 in the diagonals and 0.1 in the off-diagonal. *Block.splslda()* was used to estimate the number of components to use; the most frequent estimate of number of components (median *choice.ncomp $WeightedVote*) was used as the number of components in the

model. A grid-based search was performed to estimate the number of variables to retain for each omic layer and component. Specifically, *tune.block.splsda()* was used with 10-fold cross-validation to ascertain the number of variables to keep (between 5 and 30 variables). DIABLO was used to select variables for each component.

## Data availability

The R software implementation of netDx is available as a public GitHub repository at: https://github.com/BaderLab/netDx. A Docker image with netDx installed is made available at Zenodo (https://doi.org/10.5281/zenodo.2558452). It includes all the data and software needed to reproduce results in this manuscript.

**Expanded View** for this article is available online.

## Acknowledgments

We thank Quaid Morris and Daniele Merico for discussions on method development and Lincoln Stein for feedback on earlier versions of the manuscript. We also thank Han Liang for discussion on implementation details for the machine learning used in Yuan *et al* (2014a). This work was supported by a Canadian Institutes of Health Research award to GDB (grant number 499509), by the NRNB (U.S. National Institutes of Health grant number 499382), by a grant for Cytoscape (U.S. National Institutes of Health grant number 503758), and by a Canadian Institutes of Health Research Fellowship award to SP (award number 498002).

## Author contributions

All authors contributed to netDx method development. SP and MAS analyzed the data. SP wrote the netDx software package with contributions from MAS. SH, HK, and RI developed initial versions of netDx. SP and GDB wrote the paper. GDB supervised the work.

## Conflict of interest

The authors declare that they have no conflict of interest.

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
