## [Review Process File · Molecular Systems Biology]

netDx: Interpretable patient classification using integrated patient similarity networks

Shraddha Pai, Shirley Hui, Ruth Isserlin, Muhammad A Shah, Hussam Kaka and Gary D. Bader.

Review timeline:

Submission date:	12 th June 2018
Editorial Decision:	23 rd July 2018
Revision received:	29 th November 2018
Editorial Decision:	5 th February 2019
Revision received:	12 th February 2019
Accepted:	13 th February 2019

Editor: Maria Polychronidou

Transaction Report:

1st Editorial Decision

23rd July 2018

Thank you again for submitting your work to Molecular Systems Biology. We have now heard back from the three referees who agreed to evaluate your study. As you will see below, the reviewers think that the presented approach seems interesting. They raise however a series of concerns, which we would ask you to address in a revision.

The reviewers' recommendations are rather clear so I think that there is no need to repeat the points listed below. Importantly, methodological details and software implementation should be made available to the reviewers. Please feel free to contact me in case you would like to discuss in further detail any of the issues raised by the reviewers.

REFeree REPORTS

Reviewer #1:

The authors describe netDx, a procedure for performing supervised machine learning using similarity metrics. In this case, features are collapsed to similarity scores (usually just a Pearson correlation, going by Supplementary Table 1) before being fed into a machine learning algorithm.

Though the performance appears to be good, I also have some concerns related to the presentation. I may also have concerns about the underlying work, but I don't know that because the methods section appears to be relatively incomplete. It would be impossible for me to provide a substantive peer review on the methods/results given what is currently available.

The authors only lightly describe which exact performance metric goes into Figure 2 for netDx and how that performance is obtained. Is the variability from performance over splits, over models, etc? I need this information to make any judgement on the manuscript's validity.

Though the limited description makes it hard to assess the methods and results, I can provide some feedback on the framing. The authors describe this approach as being used for precision medicine. I have considerable numbers of concerns around this point:

* Similarity-based methods by design leak the information of participants. It's unclear how this can be mitigated. A classifier can at least be distributed, but for similarity you need to distribute enough information for someone else to calculate similarity + the clinical information.

* "netDx provides a complete framework for precision medicine." It does not appear to actually provide this. It would be better to more accurately define the scope of what netDx provides.

The checklist is both too specific to netDx and too incomplete all at the same time. It might be worth considering whether or not this would be better handled via a separate review or perspective paper. If it's going to be handled here, please consider the extent to which it is overly tuned for this context. For example "... in the integrated similarity network" appears to apply, in all the copious literature on the topic, to only this paper. Also, steps that seem relevant like independent validation in a separate cohort and prospective trials are both missing. Finally, the term orthogonal appears to be applied to something that does not appear to be orthogonal: "in the context of survival prediction in cancer, a predictor should result in significantly separable survival curves for the two predicted patient sets."

Finally, the largest problems associated with using similarity-based methods for precision medicine appear to be unaddressed. It requires a large bank of samples to be able to begin to make predictions, and it's unclear how similarity scores will generalize across all platforms. Since every value from every data type can contribute to the predictor, it may be substantially harder to deploy these methods in a production setting. This doesn't need methodological work within this exact contribution, but should be acknowledged as a limitation. As framed though, this appears to be a glaring weakness of both the method and the lack of independent test sets measured by different groups.

Minor:

* Remove the discussion of "borderline significant." Since the decision has been made to use this approach to assessing the robustness of results, either it is or it is not significant depending on the selected alpha.

Reviewer #2:

In this manuscript, the authors introduce a new method for survival-classification of patient by integration of omics and clinical data. The methodology is novel and generally well described. The evaluation adequately demonstrates that the method provides an advantage over conventional machine learning methods, and provides novel biological insights when integrated with molecular pathway information. The manuscript suffers from a number of unclear/imprecise formulations, and is missing some detailed information here and there that should be provided in the supplements. If these shortcomings are addressed, I believe the manuscript is excellent and suitable for publication.

Page 6, line 4 from bottom: You write that feature networks are integrated using "an established association network integration algorithm" and cite two papers (ref. 12 and 13), the first one being the 3Prop, and the second being an update paper for GeneMANIA. It appears the 3Prop algorithm is used, but this is not perfectly clear. I suggest mentioning 3Prop by name and/or removing ref. 13, as it is not relevant here.

The performance of NetDx is described as "excellent" in several places. While this is up to interpretation, the mean AUROCs generally vary between 0.6 and 0.8, and judging from Supp. table 1, the increase in performance from data integration seems to come mostly from the clinical data. While NetDx does perform comparatively well to the other methods, "excellent" is, to me, misleading, especially when used in the abstract without context.

Page 22, line 4 from bottom (Figure 2 caption): You write that you show a "representative split"

from the survival models trained on KIRC. Please specify what constitutes representative in this context. Given the high variance in performance of the different splits, this is a significant detail.

Page 22, Figure 2: A one-sided Mann-Whitney U-test is used for comparing netDx performance to all other methods. Please provide a justification for using a one one-tailed test instead of a two-tailed test.

It would also be beneficial to include significance tests between netDx and each method individually (in supplementary table.)

Supplementary table 1: Please include variance of AUROC values in addition to mean. Also, please include AUROC values for methods compared to as well for reference.

Online methods, page 4, page mid: "an edge weight at floating-point precision". The term "floating-point precision" is unconventional, and may be confusing to less technically-oriented readers. More common terms would be "machine epsilon" or "unit roundoff". Consider writing "an edge with the smallest possible non-zero weight" or something similar instead.

Reviewer #3:

In this manuscript, the authors report a network-based classification model with applications in multi-omics integration and personalised medicine. The mathematical framework appears to be solid and builds upon Patient Similarity Networks, an approach that has previously been applied for unsupervised clustering (Wang et al, 2014 and Lie et al, 2015). The authors extend this framework to a supervised setting, with the aim to classify patients while identifying relevant discriminatory features.

Important advantages of this method compared to conventional classification algorithms are the natural integration of different data modalities, as well as gains in interpretability by combining features into predefined gene sets that reflect biological knowledge.

Overall, the manuscript is clear, well-structured and the specific applications are of broad interest. However, we have technical concerns that would need to be addressed. In addition, we request access to the software implementation for review purposes.

Major comments:

When using netDx using single features, rather than aggregated gene sets, the model appears to be conceptually related to conventional approaches, such as logistic regression or random forest classification. Yet, the authors show in Figure 2, that the model offers practical performance gains. Can the authors provide further details where these gains come from?

The method builds on a series of filtering- and selection steps to identify the most relevant features. Given these operations, we feel that there is a risk of an implicit multiple testing, that would need to be controlled for. We would request additional controls, for example using permutations to define empirical null relationships, to assess the robustness of the approach.

In the quantitative assessment of the method (Figure 2), are the described steps to filter specific features only applied to netDx? Differences in data preprocessing and filtering hampers an objective comparison with alternative methods. It would seem important to tease apart performance differences of the actual methods versus differences in pre-processing of the data.

The authors show that the method yields interpretable classifications (Figure 3,4). However, these results are not compared to alternative methods. We would request that the author contrast these results to gene set enrichment analysis applied to the output of classical univariate methods (i.e. random forests or logistic regression), as well as methods that can cope with multi-omics data (e.g. DIABLO, implemented in the mixOmics package, Rohart et al 2017).

Specifically in the multi-omics settings, it is common that entire assays are missing for a subset of the samples. Can the model cope with this structure of missing data?

Minor comments:

Is there a stochastic component on the model? If so, can the authors show consistency statistics across multiple applications of the method?

An explanation for the asterisks in Figure 2 is missing.

netDx methods paper: MSB-18-8497

Responses are in bold under individual points raised by the reviewers.

Reviewer #1:

The authors describe netDx, a procedure for performing supervised machine learning using similarity metrics. In this case, features are collapsed to similarity scores (usually just a Pearson correlation, going by Supplementary Table 1) before being fed into a machine learning algorithm.

Though the performance appears to be good, I also have some concerns related to the presentation. I may also have concerns about the underlying work, but I don't know that because the methods section appears to be relatively incomplete. It would be impossible for me to provide a substantive peer review on the methods/results given what is currently available.

The authors only lightly describe which exact performance metric goes into Figure 2 for netDx and how that performance is obtained. Is the variability from performance over splits, over models, etc? I need this information to make any judgement on the manuscript's validity.

Thank you for this important feedback. In response to this and the feedback of other reviewers, we have added substantially more detail to the manuscript in the form of supplementary tables, figures, new sections and expanded text in the Results section. In addition, all the code and data to reproduce results in this study have been packaged as a Docker container to make it easier to see exactly how the results were generated.

In Figure 2, a box plot shows performance variation over models, where each model uses a different combination of input data types (e.g. clinical + mRNA). For each model, performance is measured using average AUROC across 20 train/test splits run on the full netDx pipeline (including feature selection). We clarified this in the caption text. Also:

- 1. We added a new Figure EV1 which shows a detailed workflow of the netDx classifier algorithm and highlights stochastic components of the method.**
- 2. To detail the comparison of the workflow of netDx and PanCancer survival analysis (processing, predictor design, univariate filtering choices), we added a new Figure EV3 and Appendix Table S1.**
- 3. To support reproducibility and review, we provide reviewers with a Docker image with netDx software installed, code for all analyses reported in this manuscript, data, pre-generated results and documentation so that figures can be reproduced (Available at: <https://tinyurl.com/y9b4alf1>). This will be part of the software distribution once netDx is publicly released.**

Though the limited description makes it hard to assess the methods and results, I can provide some feedback on the framing. The authors describe this approach as being used for precision medicine. I have considerable numbers of concerns around this point:

* Similarity-based methods by design leak the information of participants. It's unclear how this can be mitigated. A classifier can at least be distributed, but for similarity you need to distribute enough information for someone else to calculate similarity + the clinical information.

This is an excellent point. The scenario we had envisaged in terms of data sharing is that a classifier could be created as a service where Party A (classification user and patient data provider) shares a set of patient similarities and anonymous patient labels with Party B (the

classification service) and the results can be returned in terms of anonymized patient identifiers. In this case, the patient data remains private with Party A and only the similarities need to be shared with Party B. The similarity computing code would be available for Party A to use to convert their data into the similarity matrix form that Party B needs. Classification results are sent back to Party A. Party B never needs to see private patient information. This scenario is relevant to research teams who are analyzing their own patient cohorts.

However, in light of the reviewer comment, this is not relevant in a production classifier scenario, where a generally useful classifier is shared with the world while also maintaining the privacy of the training data. In this scenario, Party A creates a classifier and wants to share the classifier with Party B, but without sharing any of the patient information used to create the classifier. This of course is not possible with our current system, as pointed out, because classification would require computing similarities between new patients and those in the training data. A traditional classifier would not have this problem, as it takes as input either the raw data or a selected set of informative variables that could be used to classify new data without sharing the original training data.

We may be able to address this in a future system in a few ways:

- 1) We could develop a new method that computes similarities to the training set samples given a set of public training features (e.g. gene expression values from a publicly available data set such as The Cancer Genome Atlas). A subset of data features could be selected to reduce the need to access all the raw data.
- 2) Required training data could be shared under standard secure and ethical mechanisms, including encrypted data transfer and data access and use agreements, similar to how dbGAP or EGA works with many existing data sets.
- 3) We could use differential privacy techniques to anonymize the training data in a way that enables us to share it without enabling re-identification, but still enables using it to compute similarities (e.g. by using noise to mask training data signal and prevent identification, but still maintaining enough signal to compute similarities).
- 4) We could use distance preserving homomorphic encryption from the field of privacy-preserving data mining (e.g. to enable secure clustering - <http://insticc.org/node/TechnicalProgram/ic3k/presentationDetails/68908>).

In any case, netDx does not require data privacy issues to be solved to be useful, thus we have removed claims related to this point and instead briefly discuss the need for future work in this area.

* "netDx provides a complete framework for precision medicine." It does not appear to actually provide this. It would be better to more accurately define the scope of what netDx provides.

We have toned down this claim and now claim that it can be a useful tool for precision medicine research.

The checklist is both too specific to netDx and too incomplete all at the same time. It might be worth considering whether or not this would be better handled via a separate review or perspective paper. If it's going to be handled here, please consider the extent to which it is overly tuned for this context. For example "... in the integrated similarity network" appears to apply, in all the copious literature on the topic, to only this paper. Also, steps that seem relevant like independent validation in a separate cohort and prospective trials are both missing. Finally, the term orthogonal appears to be applied to something that does not appear to be orthogonal: "in the context of survival prediction in cancer, a predictor should result in significantly separable survival curves for the two predicted patient sets."

We have removed the predictor checklist idea from this manuscript. We agree that it could be part of a review paper.

Finally, the largest problems associated with using similarity-based methods for precision medicine appear to be unaddressed. It requires a large bank of samples to be able to begin to make predictions, and it's unclear how similarity scores will generalize across all platforms. Since every value from every data type can contribute to the predictor, it may be substantially harder to deploy these methods in a production setting. This doesn't need methodological work within this exact contribution, but should be acknowledged as a limitation. As framed though, this appears to be a glaring weakness of both the method and the lack of independent test sets measured by different groups.

This is true and also a limitation for most machine learning methods. We have acknowledged this limitation and changed the framing to clarify the more realistic scope and utility of the method.

Minor:

* Remove the discussion of "borderline significant." Since the decision has been made to use this approach to assessing the robustness of results, either it is or it is not significant depending on the selected alpha.

Done.

Reviewer #2:

In this manuscript, the authors introduce a new method for survival-classification of patient by integration of omics and clinical data. The methodology is novel and generally well described. The evaluation adequately demonstrates that the method provides an advantage over conventional machine learning methods, and provides novel biological insights when integrated with molecular pathway information. The manuscript suffers from a number of unclear/imprecise formulations, and is missing some detailed information here and there that should be provided in the supplements. If these shortcomings are addressed, I believe the manuscript is excellent and suitable for publication.

Page 6, line 4 from bottom: You write that feature networks are integrated using "an established association network integration algorithm" and cite two papers (ref. 12 and 13), the first one being the 3Prop, and the second being an update paper for GeneMANIA. It appears the 3Prop algorithm is used, but this is not perfectly clear. I suggest mentioning 3Prop by name and/or removing ref. 13, as it is not relevant here.

Apologies, the link to 3Prop was incorrect. We use the original GeneMANIA algorithm. The text has been updated to reflect this and the citation is now for the 2008 GeneMANIA algorithm paper.

The performance of NetDx is described as "excellent" in several places. While this is up to interpretation, the mean AUROCs generally vary between 0.6 and 0.8, and judging from Supp. table 1, the increase in performance from data integration seems to come mostly from the clinical data. While NetDx does perform comparatively well to the other methods, "excellent" is, to me, misleading, especially when used in the abstract without context.

The word “excellent” has now either been replaced in all the locations where it is used in the abstract and text.

Page 22, line 4 from bottom (Figure 2 caption): You write that you show a "representative split" from the survival models trained on KIRC. Please specify what constitutes representative in this context. Given the high variance in performance of the different splits, this is a significant detail.

We used a split with AUROC closest to the average AUROC for all splits. We updated Figure 2 caption accordingly.

Page 22, Figure 2: A one-sided Mann-Whitney U-test is used for comparing netDx performance to all other methods. Please provide a justification for using a one one-tailed test instead of a two-tailed test.

We used the one-tailed test to ascertain strictly higher performance of netDx relative to other methods. This has been clarified in the manuscript.

It would also be beneficial to include significance tests between netDx and each method individually (in supplementary table.)

Appendix Table S3 now shows this for each compared machine learning method. The median performance by netDx (AUROC of 0.67) exceeds that of all other methods (AUROC ranging from 0.615 for random forests to 0.655 for Nearest Centroids). Nominal p values are less than 0.05 for 6 out of 8 comparisons.

Supplementary table 1: Please include variance of AUROC values in addition to mean. Also, please include AUROC values for methods compared to as well for reference.

We have updated Appendix Table S2 (previously Supplementary Table 1) to include the variance of AUROC values.

Appendix Table S4 now has the values reproduced from the original PanCancer survival paper, including the various methods to which netDx was compared.

Online methods, page 4, page mid: "an edge weight at floating-point precision". The term "floating-point precision" is unconventional, and may be confusing to less technically-oriented readers. More common terms would be "machine epsilon" or "unit roundoff". Consider writing "an edge with the smallest possible non-zero weight" or something similar instead.

We agree and updated the relevant sentences to use “smallest possible non-zero weight”.

Reviewer #3:

In this manuscript, the authors report a network-based classification model with applications in multi-omics integration and personalised medicine. The mathematical framework appears to be solid and builds upon Patient Similarity Networks, an approach that has previously been applied for unsupervised clustering (Wang et al, 2014 and Lie et al, 2015). The authors extend this framework to a supervised setting, with the aim to classify patients while identifying relevant discriminatory features.

Important advantages of this method compared to conventional classification algorithms are the natural integration of different data modalities, as well as gains in interpretability by combining features into predefined gene sets that reflect biological knowledge.

Overall, the manuscript is clear, well-structured and the specific applications are of broad interest. However, we have technical concerns that would need to be addressed. In addition, we request access to the software implementation for review purposes.

Thanks for the positive comments. We apologize that you were not able to access the software implementation. Details for access were in the abstract of the original submission, but may have been missed because of a page break at the end of the Abstract. “(Reviewer note: download at <http://netdx.org/index.php/netdx-reviewer-page/> (password psn123). Upon publication, netDx will be made publicly available via GitHub.)”

For easier reproducibility, we have now created a Docker container with all the software and data needed to reproduce the analyses in this manuscript. The container comes with netDx installed, and a README file details the code needed to run every analysis, or plot results from pre-generated runs. These files have been made available as part of the resubmission in a shared Dropbox folder at: <https://tinyurl.com/y9b4alfl>.

Major comments:

When using netDx using single features, rather than aggregated gene sets, the model appears to be conceptually related to conventional approaches, such as logistic regression or random forest classification. Yet, the authors show in Figure 2, that the model offers practical performance gains. Can the authors provide further details where these gains come from?

Where the univariate (one network per gene) feature model performs the best, netDx does not, in fact, outperform other methods on average. The table below shows all instances where the best-performing netDx model involves univariate features; the average AUROC for netDx is 0.57 (variance=0.02), and average AUROC for other methods is 0.60 (variance=0.015), though these are not significantly different from each other (p=0.3 from two-tailed t-test).

Tumour type	Datatype	netDx best (univariate model)	Other methods. Yuan et al. (2014)
GBM	cnv	0.56	0.51
GBM	DNAm	0.58	0.59
GBM	miRNA	0.52	0.57
GBM	rna	0.59	0.59
LUSC	clinical, rna	0.66	0.67
LUSC	miRNA	0.44	0.57
LUSC	rna	0.60	0.67
OV	cnv	0.66	0.61
OV	DNAm	0.57	0.61
OV	miRNA	0.55	0.62
average		0.57	0.60

However, we can comment on the performance gain for models with aggregate features (e.g. a single feature based on all genes in transcriptomic data). For instance, for KIRC, the basic netDx model outperformed other methods without any tuning (e.g. average AUROC across all models was 0.75 for netDx and 0.70 for other methods combined). Some performance gain may be due to the use of regularized regression in GeneMANIA; the original GeneMANIA algorithm benchmarking in the context of gene function prediction found that this reduced overfitting and resulted in superior performance relative to the use of unregularized regression.

In other instances (LUSC and OV, and some GBM models), tuning the sparsification level of the input networks - e.g. keeping the strongest X% of edges per patient - changed netDx's performance sufficiently to outperform other methods. Sparsification parameters include:

- 1) How many of each patient's strongest edges to retain (i.e. top 30-50 per patient); the optimal parameter choice differed depending on the data set (described in the Methods section, "Sparsification of Input Networks")
- 2) Applying an exponential scaling filter on edge weights (described in Methods section, "PanCancer Survival Benchmark Models: Variable prefiltering and Scaled Euclidean / Pearson")

These types of parameter changes were also found to positively influence the performance of GeneMANIA in the context of gene function prediction and presumably achieves performance gains by accentuating signal and reducing noise.

Therefore, we attribute the improved performance of netDx to the fine-tuning of the input networks through sparsification and to the use of regularized regression in the GeneMANIA algorithm used by netDx.

We have added a paragraph to the discussion section of the manuscript, discussing the value of regularization and sparsification to improved classifier performance (Page 19, bottom).

The method builds on a series of filtering- and selection steps to identify the most relevant features. Given these operations, we feel that there is a risk of an implicit multiple testing, that would need to be controlled for. We would request additional controls, for example using permutations to define empirical null relationships, to assess the robustness of the approach.

All of the filtering and selection steps are part of the feature selection process, which is solely concerned with ranking features, not testing their significance for outcome prediction. Standard multiple testing correction procedures, such as FDR, would not change this ranking. We are careful that the feature selection process is applied only to training and not test samples. Additionally, we repeat this feature selection step on multiple splits of the training data, which helps assess robustness.

We performed 20 train/test data splits for each of the 40 netDx prediction models we report (performance measures stabilized at 20 splits). For models with pathway-level features (breast cancer and asthma), we chose 100 splits to ensure variance in pathway-level feature scores stabilized (Appendix Figure S2). We have clarified these points in the manuscript: 1) The role of prefiltering in ranking features (Page 20, “PanCancer Survival benchmark models”) 2) Multiple train/test splits to stabilize performance on blind test samples (Page 11, “Benchmarking performance by predicting binarized survival in cancer”), and to achieve stable pathway-level scores (Page 13, “Pathway-level feature selection identifies cellular processes predictive of clinical condition”)

In the quantitative assessment of the method (Figure 2), are the described steps to filter specific features only applied to netDx? Differences in data preprocessing and filtering hampers an objective comparison with alternative methods. It would seem important to tease apart performance differences of the actual methods versus differences in pre-processing of the data.

We now include a more detailed description of the various data preprocessing steps and differences in predictor design, and compare netDx to the PanCancer survival analysis methods in Appendix Table S1, as well as in Figure EV3. In summary:

- 1) We downloaded already processed data from the PanCancer survival project (Yuan et al.) (<https://www.synapse.org/#!Synapse:syn1710282/wiki/27303>)
- 2) We used identical variable coding to Yuan et al.
- 3) Similar to Yuan et al. we use prefiltering within a resampling feature selection loop, except that we use lasso regression (because it is an established method known to have good performance) while they use ANOVA and shrunken centroids.
- 4) Similar to Yuan et al. we use imputation within the training/resampling loop, except that we use it only for GBM and only use imputation by median (not by mode), because we found this choice sufficient to provide a performance improvement.

The authors show that the method yields interpretable classifications (Figure 3,4). However, these results are not compared to alternative methods. We would request that the author contrast these results to gene set enrichment analysis applied to the output of classical univariate methods (i.e. random forests or logistic regression), as well as methods that can cope with multi-omics data (e.g. DIABLO, implemented in the mixOmics package, Rohart et al 2017).

Specifically in the multi-omics settings, it is common that entire assays are missing for a subset of the samples. Can the model cope with this structure of missing data?

1) We added a new paragraph in the Discussion section, comparing netDx to both GSEA and DIABLO, and two accompanying supplementary figures (Appendix Figure S3 and S4), one per comparison. In summary:

- A. netDx selects qualitatively similar pathway themes as GSEA but is more conservative (selecting fewer pathways). We attribute this stricter selection of pathways to redundancy-reduction by regularization in the feature-scoring step.**
- B. We compare netDx and DIABLO by integrating RNA and miRNA for binary classification of breast tumours. Both have similar accuracy (91% for netDx and 90% for DIABLO). netDx has built-in support for pathway selection, which provides insight into cellular processes that discriminate between classes. DIABLO selects individual data variables (e.g. microRNA or mRNA expression levels) within each data type, thus is better suited for e.g. developing a panel of biomarker molecules. Unfortunately DIABLO currently has no built in functionality to easily generate pathway-level features and visualize the resulting selected features (e.g. there is nothing similar to netDx's EnrichmentMap view). We therefore conclude that DIABLO and netDx provide complementary views (i.e. pathway-level in netDx vs. molecule-level in DIABLO) of the system and could be useful in tandem.**

2) Yes, netDx can handle entire assays missing for a subset of samples. Consider a dataset with RNA, DNAm, and miRNA data. Suppose patients {A,B,C,D,E} all have RNAseq measurements. E does not have DNAm, and C,D,E don't have miRNA. During patient similarity network construction, all five patients would be used for the RNA network; only A-D would be used for the DNAmeth network; and only A-B would be used for the miRNA network. All five patients are still represented in one or more features. The GeneMANIA algorithm is still able to integrate networks, based on the available data.

Minor comments:

Is there a stochastic component on the model? If so, can the authors show consistency statistics across multiple applications of the method?

Yes, there are three stochastic components in the model; the table below lists these and how stochasticity is handled in the model as well as which consistency statistics are used. To make the stochastic components transparent to the reader, we have added a new supplementary figure that shows the detailed workflow of netDx, highlighting components with stochasticity (Figure EV1).

Stochastic component	Consistency statistics
Splitting of samples into train and blind test	We report performance over multiple train/test splits. Variability of AUROC scores are provided (reported in Appendix Supplementary Table S2).
Univariate filtering (optional). We use lasso regression, which uses stochastic resampling.	The netDx pipeline sets the random number generator seed once for each train/test split, thus univariate filtering varies per split. We have added a supplementary figure showing the frequency with which individual variables are selected by regularized

	regression, using an ovarian cancer predictor as an example (Appendix Figure S1).
Feature selection uses stochastic resampling to evaluate stability.	Stability is reflected in the scores of individual features, measured by how frequently they are selected across resamplings. Appendix Table S5 and 6 show feature scores for the breast cancer and asthma example.

An explanation for the asterisks in Figure 2 is missing.

Thanks for catching this. The asterisks had indicated instances where netDx performance was significantly better than other methods ($p < 0.05$ from a one-tailed Wilcoxon-Mann-Whitney). The asterisk has been replaced by p-values.

2nd Editorial Decision

5th February 2019

Thank you for sending us your revised manuscript. I apologize for the delay in sending you a decision, which was due to the fact that we never received a report from reviewer #3 after a series of reminders. In order to not delay the process further, we have decided to proceed with making a decision based on our evaluation of your responses to reviewers #2 and #3 and on the comments of reviewer #1. We consider the issues raised by reviewers #2 and #3 resolved by the preformed revisions and clarifications. As you will see below, reviewer #1 also thinks that the study is now suitable for publication, pending some minor text modifications, which we would ask you to perform in a minor revision.

Before we formally accept the paper for publication we would also ask you to address the following editorial issues.

Reviewer #1:

The priority claim here seems not well justified: "It is the first classifier to apply the idea of recommender systems, similar to those used in Amazon or Netflix ("find movies like this one"), to precision medicine ("find patients who don't respond to therapy")." A quick search finds the term recommender system applied to treatment prioritization from at least as early as 2014.

My other concerns around the framing have been addressed. Some revisions would make this easier to read, like picking a term and sticking with it. For example, "(we use the terms "input networks" and "features" interchangeably)" just makes the manuscript a bit harder to read. A careful editing for clarity and ease of reading would be helpful to the reader, but wouldn't require re-review.

2nd Revision - authors' response

12th February 2019

Responses to Reviewer #1

The priority claim here seems not well justified: "It is the first classifier to apply the idea of recommender systems, similar to those used in Amazon or Netflix ("find movies like this one"), to precision medicine ("find patients who don't respond to therapy")." A quick search finds the term recommender system applied to treatment prioritization from at least as early as 2014.

Response: We thank the reviewer for re-reviewing our manuscript. The claim above has been toned down. The text now reads: "[netDx] applies the idea of recommender systems, similar to those used in Amazon or Netflix ("find movies like this one"), to precision medicine ("find patients who don't respond to therapy")."

My other concerns around the framing have been addressed. Some revisions would make this easier to read, like picking a term and sticking with it. For example, "(we use the terms "input networks" and "features" interchangeably)" just makes the manuscript a bit harder to read. A careful editing for clarity and ease of reading would be helpful to the reader, but wouldn't require re-review.

Response: As per the request, we have read through the manuscript and revised the text. We now consistently use the term "feature", as it is standard machine-learning terminology and clarifies the use of networks in the context of the machine-learning algorithm. We reserve the use of the term "network" for those instances where the network properties are relevant to the context of the sentence.

Accepted

13th February 2019

Thank you again for sending us your revised manuscript. We are now satisfied with the modifications made and I am pleased to inform you that your paper has been accepted for publication.

Corresponding Author Name: Gary Bader

Manuscript Number: MSB-18-8497